# Peptide-Resorcinarene Conjugates Obtained via Click Chemistry: Synthesis and Antimicrobial Activity

**DOI:** 10.3390/antibiotics12040773

**Published:** 2023-04-18

**Authors:** Héctor Manuel Pineda-Castañeda, Mauricio Maldonado-Villamil, Claudia Marcela Parra-Giraldo, Aura Lucía Leal-Castro, Ricardo Fierro-Medina, Zuly Jenny Rivera-Monroy, Javier Eduardo García-Castañeda

**Affiliations:** 1Chemistry Department, Universidad Nacional de Colombia, Bogotá 111321, Colombia; hmpinedac@unal.edu.co (H.M.P.-C.); mmaldonadov@unal.edu.co (M.M.-V.); allealc@unal.edu.co (A.L.L.-C.); rfierrom@unal.edu.co (R.F.-M.); zjriveram@unal.edu.co (Z.J.R.-M.); 2Human Proteomics and Mycosis Unit, Infectious Diseases Research Group, Department of Microbiology, Pontificia Universidad Javeriana, Bogotá 110231, Colombia; claudia.parra@javeriana.edu.co

**Keywords:** dendrimers, resorcinarene, click chemistry, CuAAC, antibacterial activity, antifungal activity, clinical isolate

## Abstract

Antimicrobial resistance (AMR) is one of the top ten threats to public health, as reported by the World Health Organization (WHO). One of the causes of the growing AMR problem is the lack of new therapies and/or treatment agents; consequently, many infectious diseases could become uncontrollable. The need to discover new antimicrobial agents that are alternatives to the existing ones and that allow mitigating this problem has increased, due to the rapid and global expansion of AMR. Within this context, both antimicrobial peptides (AMPs) and cyclic macromolecules, such as resorcinarenes, have been proposed as alternatives to combat AMR. Resorcinarenes present multiple copies of antibacterial compounds in their structure. These conjugate molecules have exhibited antifungal and antibacterial properties and have also been used in anti-inflammatory, antineoplastic, and cardiovascular therapies, as well as being useful in drug and gene delivery systems. In this study, it was proposed to obtain conjugates that contain four copies of AMP sequences over a resorcinarene core. Specifically, obtaining (peptide)_4_-resorcinarene conjugates derived from LfcinB (20–25): RRWQWR and BF (32–34): RLLR was explored. First, the synthesis routes that allowed obtaining: (a) alkynyl-resorcinarenes and (b) peptides functionalized with the azide group were established. These precursors were used to generate (c) (peptide)_4_-resorcinarene conjugates by azide-alkyne cycloaddition CuAAC, a kind of click chemistry. Finally, the conjugates’ biological activity was evaluated: antimicrobial activity against reference strains and clinical isolates of bacteria and fungi, and the cytotoxic activity over erythrocytes, fibroblast, MCF-7, and HeLa cell lines. Our results allowed establishing a new synthetic route, based on click chemistry, for obtaining macromolecules derived from resorcinarenes functionalized with peptides. Moreover, it was possible to identify promising antimicrobial chimeric molecules that may lead to advances in the development of new therapeutic agents.

## 1. Introduction

The indiscriminate use of antibiotics has generated an increase in antimicrobial resistance (AMR). Recent reports issued by the World Health Organization (WHO) describe the importance of finding new molecules that are able to counteract AMR; this problem is considered to be a growing threat to health worldwide [1,2]. As an example, the WHO has reported bacterial and fungal strains that exhibit greater resistance. Among these are found: *Acinetobacter*, *Pseudomonas*, and several *Enterobacteriaceae* (including *Klebsiella*, *Escherichia coli*, *Serratia,* and *Proteus*), followed by *Enterococcus faecium* and *Staphylococcus aureus*, among others [3,4], as well as the appearance of drug-resistant fungi of *Candida* species (*Candida albicans*, *C. glabrata*, *C. tropicalis*, *C. parapsilosis*, and *C. Krusei*).

In recent decades, antimicrobial agents derived from protein sources have been developed, and peptides with greater antimicrobial potential against a wide variety of resistant microorganisms than their original proteins have been found. Antimicrobial peptides (AMPs) have been described as a primary defense barrier against infections caused by external pathogens [5]. Currently, AMPs are a source of the design and development of new antimicrobial molecules. To obtain therapeutic agents based on AMPs, modifications to the original sequences have been proposed, including (i) the incorporation of unnatural amino acids, (ii) sequences with conformational restriction, (iii) peptide chimeras, (iv) palindromic structures, and (v) structures with multiple presentation of the active motif; the latter may contain an antibacterial motif of peptide origin or an organic/inorganic core [6]. Obtaining polyvalent peptides can be associated with various forms synthesis pathways, e.g., the use of divergent or convergent strategies, the use of cores based on modified amino acids, or organic motifs as polyhydroxylated platforms, such as calixarenes and resorcinarenes. This latter has become a new source of molecules that are currently being studied for the generation of polyvalent molecules, thanks to its great versatility for synthesis and the reactivity provided by the hydroxyl groups and other reactive positions present in these structures.

In this way, resorcinarenes, also known as calix[4]resorcinarenes, are polyhydroxylated macrocyclic compounds derived from resorcinol, first synthesized by Baeyer et al. from aliphatic and aromatic aldehydes [7,8]. They are made up of four resorcinol rings joined by a bridging atom, usually carbon within a methylene group at positions 4 and 6, giving the formation of a cyclic structure typically represented as a truncated *cone* with an upper and lower rim. These bridging atoms are often replaced by aliphatic and/or aromatic chains, allowing the formation of conformational isomers (stereoisomerism) [9]. The versatility of these host systems stems from synthetically easy modifications either on the upper or lower rim of these macrocycles. However, core (unsubstituted) resorcinarene conformations are known to be modulated quite easily by reaction conditions when the host is synthesized [10]. Five possible conformers have been reported: (i) *crown*, (ii) *boat*, (iii) *saddle*, (iv) *chair*, and (v) *diamond* (Figure 1) [11]. Each of these conformations is possible, depending on aspects such as the position of the resorcinol units and the substituents in the methylene bridges; this gives rise to a library of molecules [11,12,13,14,15,16]. Thus, the versatility in the synthesis of polyhydroxylated platforms has allowed the incorporation of substituents, which allows the use of new strategies, such as click chemistry, for the synthesis of functionalized calixarenes/resorcinarenes.

Click chemistry is currently one of the most-used tools for the generation of complex organic molecules [17]. The advantages of using click chemistry in organic synthesis are remarkable: in many cases they occur under mild conditions, free of solvents, and with high yields and short reaction times, which makes this new strategy a viable alternative for obtaining conjugated molecules [18]. The present study highlights the use of click chemistry for the generation of functionalized resorcinarenes with several copies of an AMP, specifically the azide-alkyne cycloaddition catalyzed by copper (I) (Cu*AAC*) on polyhydroxylated platforms of the resorcinarene type [19,20]. Although their development is still limited and they are in the exploratory phase, antimicrobial peptide-resorcinarene conjugates are promising candidates for treating infections caused by drug-resistant bacteria and fungi. The resorcinarenes have demonstrated broad antibacterial and antifungal activity and have also exhibited functions useful as diagnostic tools, therapeutic agents, molecular recognition, and transport vehicles for genes and drugs [21].

Even though at present no peptide-resorcinarene conjugates with antibacterial and/or antifungal activity have been reported in clinical trial phases, there are some agents derived from dendrimer structures with other types of applications in clinical trials; for example, DEP^®^ docetaxel9 [22] is a drug used for solid tumors, including breast, prostate, and lung cancer, DEP^®^ cabazitaxel is a non-detergent version of the leading cancer drug [23], MAG-Tn3 dendrimer vaccine is used for breast cancer [24], ImDendrim is used for inoperable liver cancer [25], and OP-101 is used for X-linked adrenoleukodystrophy [26].

In this context, the present investigation studies the synthesis and characterization of peptide-resorcinarene conjugates derived from LfcinB and buforin, a strategy that will allow the generation of molecules that present several copies of the selected AMPs on an unprotected and polyhydroxylated nucleus.

## 2. Results and Discussion

For the synthesis of peptide-resorcinarene conjugates, we decided to evaluate three variables, so we divided our research into the following steps: step (1) evaluation of the synthetic routes that allow obtaining functionalized resorcinarenes with alkyne groups and the synthetic feasibility of peptides that contain an azide group in their structure; step (2) the synthetic parameters that allow the generation of the conjugated molecule peptide-resorcinarene by means of azide/alkyne cycloaddition; and step (3) the antibacterial activity of the conjugates obtained. These results will allow the identification of promising molecules that will generate advances in the development of new antimicrobial agents. The results obtained in step 1 to step 3 are shown below.

### 2.1. Synthesis and Characterization of Resorcinarene and Peptide Precursors

For generating the conjugates, first we had to obtain two precursors: (i) the resorcinarene core functionalized with alkyne groups, and (ii) peptides functionalized with azide motifs. It was proposed to attach the precursors via click chemistry. We selected the resorcinarenes as the core for the conjugates, because this kind of molecule has been the focus of great interest in the field of chemistry due to its low toxicity and the fact that it can be modified with a diversity of functional groups, generating a great variety of pharmacologically active derivatives, including compounds with antimicrobial activity against Gram-positive and Gram-negative bacteria, fungi, and parasites [27]. In this investigation, as a first step it was decided to carry out the synthesis of a nucleus derived from resorcinarene functionalized with alkyne groups on the lower rim. This nucleus was synthesized through acid-catalyzed cyclocondensation (see Figure 1) using resorcinol and an aldehyde previously functionalized with the propargyl group (see Appendix A).

We specifically synthesized the *C*-Tetra(4-(prop-2-yn-1-yloxy)phenyl)calix[4]resorcinarene (CTpH(F)), as is detailed in the methodology section. As can be seen in Figure 2a, the chromatographic profile of the reaction product shows two principal signals, at 9.6 min and 9.8 min. It has been reported that a single conformer or a conformational mixture can be obtained as a reaction product in the synthesis of this type of resorcinarene derivative [28,29]. To characterize the two principal products, a purification/separation and recrystallization process was carried out [30], and then each purified product was characterized via RP-HPLC, ^1^H-NMR, and ^13^C-NMR. A white solid was obtained, and its chromatographic profile shows a single species at 9.6 min with a purity of 96.4% (Figure 2b). For this product, the assignments of the ^1^H-NMR signals at low field (8.51–8.46 ppm) suggest the presence of protons from the different OH groups with an integration of four for each one, as well as the protons that make up the aromatic ring of resorcinol with signals between 6.50 and 6.12 ppm. This information was confirmed by means of ^13^C-NMR with signals at 101.7 and 101.6 ppm, Figure 3a. The characterization of this product suggests that it corresponds to the chair (*rctt*) conformer. The second product was obtained as a yellow solid; it exhibits 77.9% chromatographic purity, Figure 2c, and its ^1^H-NMR spectrum shows a single singlet signal at low field 8.54 ppm that integrates for 8 protons. This suggests that all of the protons of the OH groups of resorcinol are equivalent, and therefore this product should correspond to the *rccc* conformer, Figure 3b. The two resorcinarene derivatives were characterized using MALDI-TOF mass spectrometry. Table 1 shows the MS data obtained (product characterization is presented in Appendix A).

We selected two AMPs motifs: specifically, the following molecules were synthesized, purified, and characterized: (i) two linear peptides that correspond to the peptide sequences selected, the minimal antibacterial motifs of the AMPs LfcinB, residues 20–25: RRWQWR, and buforin BF, residues 32–35: RLLR, which were obtained to be used as control; and (ii) two azide-peptides, in which the azide group was introduced at the N-terminal end using the residue Fmoc-azidolysine, i.e., the sequences obtained were: K(N_3_)-RRWQWR-NH_2_ and K(N_3_)-RLLR-NH_2_. All the peptides were obtained via solid-phase peptide synthesis (SPPS) over Rink amide resin, using the Fmoc/tBu strategy (in Appendix A presents an azide-peptide synthesis diagram). Our results (Table 1) show that the implemented methodology was adequate for obtaining the proposed peptides. Specifically, in this investigation, an optimized methodology was used where the concentration of the Fmoc removal reagent, 4-methylpiperidine, was 2.5% *v*/*v* in DMF. It should be noted that this concentration is ten times lower than that commonly used in the reported protocols for SPPS. For most amino acids, only one coupling cycle was required for their incorporation into the peptide sequence. The TFA/water/TIPS/EDT cleavage cocktail (93/2/2.5/2.5 *v*/*v*/*v*/*v*) was adequate for control peptides; however, for azide-functionalized peptides, the EDT and TIPS scavengers were suppressed, in order to avoid the reduction of the azide motif to an amine group [31].

### 2.2. Synthesis and Characterization of Peptide-Resorcinarene Conjugates

Once the nucleus derived from resorcinarene and the peptide sequences had been successfully functionalized, the next step was to obtain the peptide-resorcinarene conjugates by means of copper-catalyzed azide-alkyne cycloaddition reaction, Cu*AAC*. Our aim was to achieve a complete functionalization of the resorcinarene, thus generating a tetravalent molecule, i.e., (peptide)_4_-resorcinarene, as is shown in Figure 2.

First, we optimized the Cu*AAC*, azide-alkyne cycloaddition conditions. As an example, Figure 2 shows the reaction between the alkynyl-resorcinarene, CTpH(F), and the azide-peptide, K(N_3_)-RLLR. Parameters such as solvent, equivalent excess, and temperature, among others, were optimized one by one. The reaction was monitored through RP-HPLC, and aliquots at different reaction times were taken. As an example, in Figure 4, the chromatographic reaction profile is shown at times 0 and 1.5 h. At time 0 (Figure 4a, t = 0 h), signals corresponding to peptide K(N_3_)-RLLR and the CTpH(F) were observed at t_R_ of 5.3 and 9.6 min, respectively. Additionally, four new signals were also observed, specifically at 6.0, 6.3, 7.2, and 7.9 min. After 1.5 h of reaction (Figure 4a, t = 1.5 h), the signal at t_R_ of 6.0 min is the major component. This result suggests that the signals between 6.3 and 7.9 min could correspond to incomplete functionalization, i.e., CTpH(F) conjugated with one, two, or three peptide chains.

The obtained products were purified by means of RP-SPE to remove the excess of catalyst, solvent, and starting reagents that did not react. The chromatographic profile of the pure CTpH(F)-(*AAC*-(K)-RLLR)_4_ (*rctt*) conjugate is shown in Figure 4b, as well as its characterization by MALDI-TOF and HR mass spectrometry, where the *m*/*z* ratio corresponds to [M+H]^+^ or [M+4H]^4+^ of the tetra-functionalized conjugated, respectively. In general, in Table 1, a summary of the characterization data for precursors and the conjugates is presented. At Appendix A present additional information about characterization of final conjugates.

### 2.3. Antibacterial/Antifungal Activity against Reference Strains and Clinical Isolates

The antibacterial activity of tetravalent conjugates (resorcinarene-peptide) was evaluated against the reference strain *E. coli* ATCC 25922, susceptible to ciprofloxacin. Resorcinareno-peptide conjugates exhibited greater antibacterial activity against the ATCC strain of *E. coli* 25922 (MICs ranging from 13 to 52 μM) than control peptides RLLR and RRWQWR. Interestingly, conjugates that contained four copies of RLLR, CT*p*H(F)-(*AAC*-(K)-RLLR)_4_ exhibited differences in their activity; specifically, the chair conformer (*rctt*) was four times more active than the crown conformer (*rccc*). These results suggest that the resorcinarene conformation directly influences the antibacterial activity. On the other hand, the conjugate that contained four copies of the sequence RRWQWR in chair conformation (*rctt*) was active against *E. coli* 25922, six times greater than the lineal sequence. The conjugate CT*p*H(F)-(*AAC*-(K)-RLLR)_4_ (*rctt*) also exhibited antibacterial effect against *S. aureus* ATCC 25923 (MIC of 13 µM). This strain is sensitive to vancomycin. Similarly, the control peptides exhibited no antibacterial effect against this strain at the concentrations tested (Table 2).

The antimicrobial activity of the conjugate CTpH(F)-(*AAC*-(K)-RLLR)_4_ (*rctt*) against clinical isolates of *E. coli*, *S. aureus*, *C. albicans*, or *C. auris* (sensitive or resistant to different antibiotics) was evaluated (Table 2). Our result showed that all the clinical isolates evaluated were sensitive to the conjugate. The conjugate resorcinarene-peptide showed the highest antibacterial effect against clinical isolated multidrug-resistant *E. coli* 301755 and clinical isolated resistant *S. aureus* 124653 (MICs values of 13 µM), the same MIC value observed for the ATCC strains. This conjugate also exhibited antibacterial activity against the other clinical isolates of *E. coli* and *S. aureus*, with MICs values ranging from 26 to 52 µM. On the other hand, the antifungal activity of conjugate CTpH(F)-(*AAC*-(K)-RLLR)_4_ (*rctt*) was tested against clinical isolates of *C. albicans* and *C. auris*. The fluconazole-resistant clinical isolate, *C. albicans* 256 HUSI-PUJ, was more sensitive to the conjugate resorcinarene-peptide (MIC 13 µM). The conjugate also exhibited antifungal activity against sensitive strains *C. albicans* SC5314 and *C. auris* (MIC of 26 µM).

The antifungal activity of the peptide RLLRRLLR, which contains two copies of the buforin minimal antibacterial motif, did not exhibit activity against the *C. albicans* strain ATCC SC5314 (sensitive to fluconazole) and clinical isolate *C. albicans* 256 HUSI- PUJ resistant to fluconazole (MIC value of >183 µM) [32], suggesting that the conjugation of the RLLR sequence with the resorcinarene motif increased the antifungal activity (Table 2). Our results suggest that the binding of the resorcinarene motif to the peptide sequence increased antifungal and antibacterial activity against reference strains and resistant and multiresistant clinical isolates. This suggests that functionalization of resorcinarene with a short cationic peptide (RLLR or RRWQWR) is a promising strategy for obtaining molecules with antibacterial potential. The incorporation of the peptide sequence into resorcinarene increased the solubility of the resorcinarene motif, allowing it to interact with the pathogenic cell. Additionally, the resorcinarene motif may increase the stability of the peptide sequence by facilitating the interaction of the cationic sequence with the cell surface.

The growth Inhibition kinetics caused by CTpH(F)-(*AAC*-(K)-RLLR)_4_ (*rctt*) was evaluated against *E. coli* ATCC 25922 (Figure 4a) for 48 h. At MIC value (13 µM), a bacteriostatic effect was observed (green line), while a bactericide effect was observed for conjugate concentrations of 26 and 52 µM (two and four times the MIC value, respectively, Figure 5). For *S. aureus* ATCC 25923, only a bacteriostatic effect was exhibited at a concentration of 52 µM (2 × MIC). These results confirm the antibacterial effect of the CTpH(F)-(*AAC*-(K)-RLLR)_4_ (*rctt*) conjugate.

### 2.4. Cytotoxic/Hemolytic Activity

The hemolytic effect of the (peptide)_4_-resorcinarene conjugates CTpH(F)-(*AAC*-(K)-RLLR)_4_ (*rctt*) and CTpH(F)-(*AAC*-(K)-RRWQWR)_4_ (*rctt*) was evaluated using concentrations from 3.1 to 200 µg/mL (Figure 6). Neither of the two conjugates exhibited a significant hemolytic effect at the MIC values observed in the antifungal and antibacterial assays. The CTpH(F)-(AAC-(K)-RLLR)_4_ (*rctt*) conjugate exhibited a low hemolytic effect (about 4% at all concentrations evaluated), suggesting that this molecule is selective for the fungal and bacterial strains evaluated. The hemolytic activity of the peptide RLLR was 63% at 200 µg/mL, while RRWQWR showed hemolysis of 1% [33]. We can infer that the generation of the tetravalent conjugates decreased the hemolytic activity of the RLLR sequence.

Our results showed that the resorcinarene-peptide CTpH(F)-*AAC*-(K(N_3_)-RLLR)_4_ (*rctt*) exhibited a lower hemolytic effect and higher antibacterial activity than the LRRL peptide, suggesting that this synthetic strategy is useful for developing treatments against bacterial infections.

Finally, the cytotoxic effect of the conjugates over fibroblasts, MCF-7, and HeLa cell lines was evaluated (Figure 7). The results show that neither conjugate exhibits significant cytotoxic effect in these models, suggesting that they are selective for the fungal and bacterial strains evaluated. Conversely, the peptide ((RRWQWR)_4_-K_2_-C_2_), which contains four copies of the RRWQWR sequence, exhibited cytotoxicity against these cancer cell lines, bacterial, and fungal strains, suggesting that the resorcinarene-peptide conjugates only affect the bacterial and fungal cells [34].

## 3. Materials and Methods

### 3.1. General Method

Reagents Rink amide resin, Fmoc-Leu-OH, Fmoc-Arg(Pbf)-OH, Fmoc-Trp(Boc)-OH, Fmoc-Gln(Trt))-OH, Dicyclohexylcarbodiimide (DCC), and 1-Hydroxy-6-chlorobenzotriazole (6-Cl-HOBt) were purchased from AAPPTec (Louisville, KY, USA). Reagents acetonitrile (ACN), trifluoroacetic acid (TFA), dichloromethane (DCM), diisopropylethylamine (DIPEA), N,N-dimethylformamide (DMF), ethanedithiol (EDT), isopropanol (IPA), methanol, and triisopropylsilane (TIS) were acquired from Merck (Darmstadt, Germany). SPE Supelclean™ LC-18 columns were purchased from Sigma-Aldrich (St. Louis, MO, USA). Trypticase Soy Agar (TSA), Mueller Hinton Agar (MHA), and Plate Count Agar (PCA) media were purchased from Scharlau. Mueller Hinton Broth medium was purchased from Merck. The *Escherichia coli* ATCC 25922 bacterial strain was purchased from the ATCC. The MCF-7 and HeLa cell lines were obtained from ATCC^®^ (Manassas, VA, USA). All the reagents were used directly, without previous purification.

^1^H spectra were recorded at 400 MHz on a Bruker Advance 400 instrument. RP-HPLC analyzes were performed on a Chomolith C18 column (Merck, Kenilworth, NJ, USA, 50 mm), using an Agilent 1200 Liquid Chromatograph (Agilent, Omaha, NE, USA). The products were analyzed on a Bruker Impact 2 LC Q-TOF MS equipped with electrospray ionization (ESI) in positive mode and on a Bruker MicroFlex MALDI-TOF mass spectrometer.

### 3.2. Peptide Synthesis

Two azide-functionalized peptide sequences derived from LfcinB (20–25) and BF (32–35) were tested, as shown in Table 1. Additionally, control peptides corresponding to the LfcinB minimal motifs were synthesized and evaluated.

The designed peptides were obtained manually using the solid-phase peptide synthesis methodology, with the Fmoc/tBu strategy (SPPS-Fmoc/tBu). Rink Amida resin (200 mg, 0.46 meq/g) was used as a solid support, which was conditioned with DMF for 12 h. (i) Removal of the Fmoc group was performed by treatment with 5% 4-methylpiperidine in DMF, 10 min at room temperature (RT) (×2); then the resin was washed with DMF (×5), IPA (3), and DCM (×3). Thereupon, a fraction of the dry resin was subjected to the Kaiser test. (ii) The Fmoc-amino acid was mixed with DCC/6-Cl-HOBt in DMF to activate it by ester formation. This reaction was stirred for 15 min at room temperature. Then, the reaction mixture was added to the deprotected resin and left to react for 1 h at RT. The solution was then discarded, the resin was washed with DMF (×3; 1 min), DCM (×3; 1 min), and the Kaiser test was performed. When this test was positive, the resin was treated again with the Fmoc-activated amino acid until the test was negative. (iii) Deprotection of the side chains and cleavage of the peptide from the resin was performed by treating the dried resin-peptide with the cleavage solution containing TFA/water/TIS/EDT (93/2/2.5/2.5% *v*/*v*/*v*/*v*), for 4 h at RT. For azide-peptides, TIS and EDT were avoided. Then, the crude peptide was precipitated by treatment with cold ethyl ether, washings (×5) were carried out with this solvent, and the solid was dried at RT. Finally, the products were analyzed via RP-HPLC.

### 3.3. Synthesis of Resorcinarenes

#### 3.3.1. Synthesis of Precursor 1

The synthesis of 4-(prop-2-yn-1-yloxy)benzaldehyde (precursor 1) was adapted from de Kivrak et al. [35]. 4-Hydroxybenzaldehyde (0.032 mmol) was dissolved in DMF. Then, 3-bromoprop-1-yne (0.042 mmol) and potassium carbonate (0.042 mmol) were added at room temperature. The resulting mixture was stirred constantly at room temperature for 4 h. After the completion of the reaction, the reaction mixture was cooled to 0 °C, filtered and washed with distilled water (yield 88.0%). Precursor 1 was characterized by means of ^1^H and ^13^C NMR, RP-HPLC and mass spectrometry. 4-(prop-2-yn-1-yloxy)benzaldehyde: white solid in yield 88.0%. M.p 76–78 °C. ^1^H-NMR, δ (DMSO-d6, room temperature, ppm): 10.08 (s, 1H, CHO), 8.05–80.3 (d, 2H aromatic CH), 7.45–7.28 (d, 2H aromatic CH), 4.96 (s, 2H, CH_2_), 2.76 (s, 1H, HC≡C).

#### 3.3.2. *C*-Tetra(4-(prop-2-yn-1-yloxy)phenyl)calix[4]resorcinarene

(10.0 mmol) of resorcinol was dissolved in (10.0 mmol) of the previously synthesized aldehyde (Precursor 1: 4-(prop-2-yn-1-yloxy)benzaldehyde). Subsequently, 25 mL of chloroform was added in a cold bath. Once the mixture was obtained, TFA (5.0 mL) was slowly added. This mixture was stirred at 60–65 °C for 32 h in an inert atmosphere. After the formation of a white precipitate, it was washed with successive washes of diethyl ether and acetone and dried with a drying gun (yield 58.9%). The solid obtained was characterized by means of ^1^H and ^13^C NMR, RP-HPLC and mass spectrometry. *C*-Tetra(4-(prop-2-yn-1-yloxy)phenyl)calix[4]resorcinarene (crown-*rccc*): white solid in yield 20.8%. M.p. > 250 °C decomposition. ^1^H NMR, δ (DMSO-d6, room temperature, ppm): δ 3.57 (t, 4H, H^7^), 4.73 (d, 8H, H^6^), 5.56 (s, 4H, H^5^), 5.52 (s, 4H, H^4^), 6.50 (s, 4H, H^3^), 6.63 (br. m, 16H, H^2^), 8.54 (s, 8H, OH). ^13^C NMR (150.9 MHz, DMSO-d6, 30 °C): δ 40.58 (s, C^12^), 55.29 (s, C^11^), 77.92 (s, C^10^), 79.72 (s, C^9^), 102.04 (s, C^8^), 113.46 (s, C^7^), 120.64 (s, C^6^), 120.64 (s, C^6^), 131.65 (s, C^5^), 138.68 (s, C^3^), 152.43 (s, C^2^), 154.58 (s, C^1^). *C*-Tetra(4-(prop-2-yn-1-yloxy)phenyl)calix[4]resorcinarene (chair-*rctt*): white solid in yield 58.9%. M.p. > 250 °C decomposition. ^1^H NMR, δ (DMSO-d6, room temperature, ppm): δ 3.50 (t, 4H, H^1^), 4.62 (d, 8H, H^2^), 5.47 (s, 4H, H^10^), 5.51 (s, 2H, H^3^), 6.12 (s, 2H, H^4^), 6.27 (s, 2H, H^5^), 6.30 (s, 2H, H^6^), 6.50 (br. m, 16H, H^7^), 8.44 (s, 4H, OH^8^), 8.51 (s, 4H, OH^9^). ^13^C NMR (150.9 MHz, DMSO-d6, 30 °C): δ 41.26 (s, C^1^), 55.29 (s, C^2^), 77.79 (s, C^3^), 79.67 (s, C^4^), 101.67 (s, C^5^), 101.70 (s, C^6^), 113.24 (s, C^7^), 120.87 (s, C^8^), 121.16 (s, C^9^), 129.34 (s, C^10^), 129.70 (s, C^11^), 131.85 (s, C^12^), 137.17 (s, C^13^), 152.44 (s, C^14^), 152.60 (s, C^15^), 154.55 (s, C^16^).

### 3.4. (Peptide)_4_-Resorcinarene Conjugate Synthesis (Click Chemistry)

For the synthesis of the dendrimers, protocols from the literature [33,36,37,38] were adapted. Briefly, the crude peptide functionalized with azide (0.05 mmol) and the resorcinarene base (0.01 mmol) were dissolved and mixed in H_2_O/DMF (1:3 *v*/*v*), and then the catalyst copper sulfate pentahydrate (CuSO_4_·5H_2_O) (0.00008 mmol) and sodium ascorbate (0.000016 mmol) were added to the reaction mixture. The mixture was maintained with constant stirring at RT, 30 °C, for 6 h. The progress of the reaction was monitored by RP-HPLC. Finally, the dendrimer was purified using RP-SPE and characterized via liquid chromatography (RP-HPLC) and mass spectrometry (MALDI-TOF).

### 3.5. Purification and Characterization

For RP-SPE, Supelclean ENVI-18 SPE cartridges (5 g bed weight, 20 mL volume) were used. SPE columns were activated before use with 30 mL methanol, 30 mL ACN (containing 0.1% TFA, solvent B) and equilibrated with 30 mL water (containing 0.1% TFA, solvent B), solvent A). Elution of the peptide or peptide-resorcinarene dendrimer was performed by increasing the percentage of solvent B in the eluent. The collected fractions were analyzed via RP-HPLC. Fractions containing the pure product were pooled and then lyophilized, and the pure product was analyzed via MALDI-TOF MS mass spectrometry. This analysis was performed on a Bruker MicroFlex MALDI-TOF mass spectrometer. RP-HPLC analyses were performed on a Chromolith C18 (50 × 4.6 mm) using an Agilent 1200 Liquid Chromatograph (Agilent, Omaha, NE, USA). Gradient elution from solvent B (0.05% TFA in acetonitrile) to solvent A (0.05% TFA in water) was performed as follows: 5/5/100/100/5/5% B at 01/01/18/21/21.1/24 min. Detection was performed at 210 nm, and the flow rate was 2 mL/min. The sample concentration was 1.0 mg/mL, and 10 µL was injected.

### 3.6. Activity Assays

#### 3.6.1. Antibacterial Activity

The minimum inhibitory concentration (MIC) was determined using a broth microdilution assay. Specifically, 90 μL of Mueller Hinton broth and 90 μL of peptide (initial concentration 444 μg/mL) were added to a 96-well plate, with serial dilutions (200, 100, 50, 25, 12.5, and 6, 2 μg/mL). Then, 10 μL (5 × 10^5^ CFU/mL) of inoculum was added to each well, and the final volume in each well was 100 μL. They were incubated for 24 h at 37 °C, and absorbance at 620 nm was measured using the ELISA reader. To determine the minimum bactericidal concentration (MBC), an aliquot was taken from each well and placed on a Mueller Hinton Agar plate. After 24 h of incubation at 37 °C, the MBC was determined. Each of these tests was performed in duplicate [39].

#### 3.6.2. Antifungal Activity

Antifungal susceptibility testing was performed using the broth microdilution (BMD) method, following CLSI M27-A3 guidelines with slight modifications [40]. Briefly, yeast suspension was prepared in 0.85% saline (SS) and adjusted to 1–5 × 10^6^ cells/mL (0.5 McFarland standard). It was then diluted in RPMI 1640 liquid medium (Sigma-Aldrich, Saint Louis, MA, USA) (with MOPS, pH 7.2) and adjusted to 0.5 × 103–2.5 × 10^3^ cells/mL. A measure of 100 µL of yeast inoculum was added to a 96-well plate containing serial dilutions of the dendrimer. The final concentrations used were 200, 100, 50, 25, 12.5, and 6.2 µg/mL. FLC drug was used as a control (0.125 to 128 µg/mL). Minimal inhibitory concentrations (MICs) were visualized and densitometry (595 nm, microplate reader, iMarKTM, Bio-rad) was used to determine the lowest concentration that caused a significant decrease compared to the untreated growth control after 48 h of incubation. The MIC endpoint was defined as the lowest concentration capable of inhibiting 80% of cell growth compared to its respective positive control. Three independent trials were performed. To verify that the tested molecule could kill yeast cells, the plates were also tested for minimal fungicidal concentration (MFC). Briefly, aliquots from each well of the susceptibility test assays were transferred to plates containing Sabouraud Dextrose Agar (SDA), which were then incubated.

### 3.7. Time-Kill Curve

The time-kill curve was constructed using the CLSI protocol, with some modifications [41]. Before testing, the bacterial strains of *E. coli* ATCC 25922 and *S. aureus* ATCC were subcultured in Tryptic Soy Agar. The colonies of a 24 h culture were suspended in 9 mL of heart infusion broth and adjusted to the standard by means of a calibration curve. These procedures resulted in an initial inoculum of approximately 5 × 10^5^ CFU/mL. The final working volumes in the peptide (0.5× MIC, MIC, and 2× MIC final concentrations) and inoculum experiments were 270 µL and 30 µL, respectively. The samples were incubated on Bioscreen C equipment for 48 h at 37 °C, and absorbance readings (600 nm) were obtained at 0 h (before the addition of peptide) and then repeatedly every hour up to 48 h.

### 3.8. Cytotoxic Activity

#### 3.8.1. Cell Culture

For all cell lines, the medium used was Dulbecco’s Modified Eagle’s Medium (DMEM)/Ham F-12 Nutrient Mix. For the MCF-7 line, the medium was supplemented with 10% fetal bovine serum (SFB), 1.5 g/L of NaHCO_3_ and NaOH up to pH 7.4, amphotericin (200 μg/mL), and 1% penicillin and streptomycin. For fibroblast cells, in addition to the above, hydrocortisone (250 µg/mL) was added. All the media were filtered through a 0.22 μm membrane.

#### 3.8.2. MTT Assay

Briefly, cells were seeded with complete medium in 96-well plates at a rate of 10,000 cells and 100 µL per well and allowed to adhere to the plates for 24 h. Subsequently, the complete medium was removed, and incomplete medium was added for synchronization for another 24 h. The cells were incubated at 37 °C for 2, 24, or 48 h with 100 μL of peptide at the concentrations to be evaluated (200, 100, 50, 25, 12.5, 6.25, and 3.1 μg/mL). Next, the peptide was removed from the box and 100 µL of incomplete medium with 10% 3-4,5-dimethylthiazol-2-yl-2,5-diphenyltetrazole (MTT) bromide was added and incubated for 4 h. The medium was replaced with 100 μL of isopropanol (IPA), and after 30 min of incubation at 37 °C, the absorbance at 575 nm was measured. Incomplete culture medium with 10% MTT was used as a negative control, and cells without MTT treatment were used as a positive control [34].

#### 3.8.3. Hemolysis Assay

First, 5.0 mL of heparinized peripheral blood was centrifuged at 1000 rpm for 7 min. The erythrocyte fraction was suspended in 10 µof saline solution (SS) and washed twice by centrifugation at 1000 rpm for 7 min. The erythrocytes (2% hematocrit) were incubated with peptide (ranging from 6.2 to 200 µg/mL) for 2 h at 37 °C. SS was used as a negative control, while distilled water was used as a positive control. The mixtures were centrifuged, the supernatants were collected, and the absorbance was determined to be 540 nm [42].

## 4. Conclusions

Using solid phase peptide synthesis (SPPS), it was possible to obtain two azide-functionalized peptides derived from LfcinB and BF. Moreover, an alkyne-functionalized resorcinarene-derived core was synthesized. Through click chemistry, Cu*AAC*, it was possible to obtain two tetra-functionalized dendrimers characterized using RP-HPLC, ESI-MS, and MALDI-TOF. These new conjugates showed enhanced antimicrobial activity against *E. coli* ATCC 25922, *S. aureus* ATCC 25923, and bacterial clinical isolates and strains of *Candida* spp. There was no evidence of any type of effect on the cytotoxic activity in blood, fibroblasts, or cancer cell lines, which means that these tetravalent (peptide)n-resorcinarene conjugates are selective antimicrobial agents. In addition, a new route for the synthesis of functionalized conjugates with peptide sequences was established, since this is the first time that the synthesis of tetra-functionalized resorcinarenes with sequences derived from LfcinB and BF has been reported.

## Data Availability

Not applicable.

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
