# Peer review of "Peptide-Resorcinarene Conjugates Obtained via Click Chemistry: Synthesis and Antimicrobial Activity"

_antibiotics, 2023, doi:10.3390/antibiotics12040773_

Round 1

Reviewer 2 Report

In manuscript “Peptide-Resorcinarene Conjugates Obtained via Click Chemistry: Synthesis and Antimicrobial Activity,” authors describe the coupling of azide-peptides derived from buforin BF residues with an alkynyl resorcinarene through CuAAC reaction. In addition, synthesized conjugates displayed an outstanding activity against yeasts of E. coli and S. aureus among others. I recommend the publication of this manuscript in Antibiotics after an additional revision.

 -Authors should consider a change of 2-D representation for structure of C-Tetra(4-(prop-2-yn-1-yloxy)phenyl)calix[4]resorcinarene in Diagram 1.  A more appropriate 3-D structure showing both “chair” and “crown” conformers should be placed instead.

-In pages 4 and 5 authors should provide a diagram/scheme about the synthesis of azide-peptide derivatives, as well as the corresponding references.

Reviewer 3 Report

The manuscript entitled "Peptide-Resorcinarene Conjugates Obtained via Click Chemistry: Synthesis and Antimicrobial Activity" by Héctor Manuel Pineda-Castañeda et al. describe:

1) the synthesis of C-Tetra(4-(prop-2-yn-1-yloxy)phenyl)calix[4]resorcinarene obtained as a mixture of two conformes (rctt + rccc)

2) separation and characterization of the two conformers

3) synthesis of peptide

4) formation and purification of peptide-resorcinarene conjugates by click chemistry

5) biological evaluation

In my opinion, the research work is well designed but, before publication on Antibiotics, some issues need to be investigated further.

a) The experimental procedure for the obtainement of C-Tetra(4-(prop-2-yn-1-yloxy)phenyl)calix[4]resorcinarene is identical to that reported by Knyazeva et al. (ref. 28) but, unlike reference 28, the authors of this article obtain a mixture of conformers. I think this different experimental result deserves extra comments.

b) Apart compound CTpH(F)-(AAC-(K)-RLLR)4 (rctt), the purity of the other two tested compounds is low (under 90% by HPLC). This could significantly affect the biological outcome. Moreover, the nature and structure of the impurities is not disclosed.

c) The yields of the click chemistry and of the purified peptide-resorcinarene conjugates are completely missing.

d) The NMR analysis of peptide-resorcinarene conjugates could by very useful in order to understand if the conformation of the starting resorcinarene core is retained in the final product.

e) I suggest that authors create Supplementary Materials containing NMR, MS, and chromatograms.

Round 2

Reviewer 3 Report

The authors replied to my comments and made the necessary changes to the manuscript. I consider the current version suitable for publication in Antibiotics